# Lifestyle Intervention to Promote an Adequate Gestational Weight Gain and Improve Perinatal Outcomes in a Cohort of Obese Women

**DOI:** 10.3390/nu16193261

**Published:** 2024-09-26

**Authors:** Daniela Menichini, Eleonora Spelta, Francesca Monari, Elisabetta Petrella, Fabio Facchinetti, Isabella Neri

**Affiliations:** 1Obstetrics Unit, Mother Infant Department, University Hospital Policlinico of Modena, 41125 Modena, Italy; speltaeleonora@gmail.com (E.S.); francesca.monari@unimore.it (F.M.); petrella.elisabetta@gmail.com (E.P.); fabio.facchinetti@unimore.it (F.F.); isabella.neri@unimore.it (I.N.); 2School of Midwifery, Department of Medical and Surgical Sciences, University of Modena and Reggio Emilia, 41125 Modena, Italy

**Keywords:** obesity, insufficient GWG, pregnancy, lifestyle intervention

## Abstract

Objective: This study aims to evaluate the correlation of gestational weight gain (GWG) with pregnancy and perinatal outcomes in a cohort of obese women class I-III receiving standard care (SC) or lifestyle intervention (LI). Methods: This is a prospective cohort study including singleton obese women (body mass index, BMI ≥ 30) who delivered between 2016 and 2020. Women exposed to a LI were referred to an obesity weight management ad hoc clinic. Women followed by family centers or private settings represented the SC group. The LI started between the 9 and 12th week, consisting of a low-calorie diet and physical activity program. Pregnancy and perinatal outcomes were prospectively collected. Women included in the SC group were followed, simply checking their pregnancy and health status, providing general recommendations on a healthy lifestyle in pregnancy. GWG was categorized as insufficient, adequate, or excessive according to the Institute of Medicine (IOM). Results: A total of 1874 obese singleton women delivered in the study period. Among them, 565 (30.1%) were included in the LI while 1309 received SC. Women in SC showed a higher rate of GWG out of the IOM recommendations (excessive/insufficient), while women in the LI group showed higher adequate GWG. The small-for-gestational-age (SGA) rate resulted to be higher in the SC group. Once adjusting for age, BMI, country of origin, provider, and gestational hypertension, the risk for SGA was increased by insufficient GWG (OR = 1.25; 95%CI: 1.03–1.59), while it was reduced by LI (OR = 0.67, 95%CI: 0.42–0.98). Conclusions: In a cohort of obese women, the exposure to an LI was associated with more adequate GWG, reduced insufficient weight gain, and a decreased risk of SGA infants.

## 1. Introduction

Obesity has become a serious health concern worldwide. In pregnancy, the metabolic changes and the pro-inflammatory chronic status that characterize obesity are associated with several health complications and adverse outcomes for both the mother and the fetus, such as preeclampsia, gestational diabetes, preterm birth, and fetal growth restriction.

In obese women, the weight that should be gained during gestation is a matter of debate, also because the guidelines of the Institute of Medicine (IOM) indicated a range of 5–9 Kg, irrespective of obesity classes [1]. The American College for Obstetrics and Gynecology (ACOG) Committee stated that an obese pregnant woman carrying an appropriate growing fetus could gain less weight than the ranges recommended by the IOM [2].

However, a systematic review demonstrated that obese women with a gestational weight gain (GWG) below the IOM guidelines (but no weight loss) showed a higher rate of preterm labor and small-for-gestational-age neonates (SGA) along with a decreased rate of large-for-gestational-age (LGA) neonates compared to those gaining within the recommended ranges [3]. A large cohort study performed by Bodnar et al. defined the ranges of weight gain associated with a low probability (<10%) of either SGA or LGA, and a minimal risk of preterm birth. The purposed ranges are as follows: 9.1–13.5 kg for obese class I, 5–9 kg for obese class II, and <5 (white woman) or <2.2 (black women) for obese class III [4]. In a Swedish population, obese women class II–III with lower-than-recommended or no GWG showed a decreased risk of many unfavorable outcomes such as LGA, cesarean section, excessive post-partum bleeding, and instrumental delivery. However, an increased risk of SGA was found in obese class III [5].

In daily practice, although not recommended by guidelines, often obese pregnant women try to restrict weight gain or even lose weight by using a self-chosen diet [6].

A retrospective cohort study in a German population reported that weight loss in overweight and obese women was associated with a reduction in pregnancy complications such as preeclampsia and non-elective cesarean section. However, apart from obese class III women, an increased risk of preterm delivery and SGA has been reported [7].

A review of studies performed in Canadian populations reported that women experiencing weight loss, in addition to reduce LGA and macrosomia, showed higher odds of SGA (either <10th or <3rd percentiles) [8]. However, the studies published so far do not consider lifestyle interventions, whether they are recommended by doctors or adopted by women themselves.

Accordingly, in a cohort of obese women class I-III scheduled to receive standard care or lifestyle intervention (low-caloric diet and physical activity) (LI), we aimed to evaluate the correlation of GWG with pregnancy and perinatal outcomes.

## 2. Materials and Methods

This prospective cohort study was approved by the Ethics Committee of Modena, Area Vasta Emilia Nord (AVEN) in April 2016, reference number 136/15. A written informed consent was collected from all the women included in the study.

The inclusion criteria for the study were as follows: having a BMI ≥ 30 kg/m^2^, age ≥ 18 years, single pregnancy, GA ≤ less than 12 weeks, and willingness to participate in the study.

The exclusion criteria were mainly represented by the presence of chronic diseases, in particular pregestational hypertension, type 1 or 2 diabetes mellitus, cardiovascular pathologies and kidney disease that contraindicate caloric restriction or physical activity, thyroid pathologies not compensated by therapy, inability to follow the diet prescribed for cultural, and ethnicity or religion-linked reasons.

The study included singleton pregnant women with a pre-pregnancy BMI ≥ 30 kg/m^2^ and age ≥ 18 years who delivered from 2016 to 2020 at the Policlinic Hospital of Modena. Among them, the family centers’ physicians and midwives proposed to a group of women the possibility to be referred to an obesity weight management ad hoc clinic at the Obstetric Unit at the Mother–Infant Department. The women that agreed represented the LI group, and were exposed, starting from their 9th–12th weeks of pregnancy, to a personalized dietary intervention tailored to their eating habits, taste, and religion, with extensive explanation of meal subdivision and possible food substitutions, and to a physical activity program.

The women in the SC group were routinely followed by family centers or private practitioners and received a simple nutritional booklet regarding lifestyle, which was in accordance with the Italian Guidelines for a healthy diet and physical activity during pregnancy [9,10].

### 2.1. Lifestyle Intervention

The LI consisted of the prescription of a low-glycemic, low-saturatedfat diet with a total intake of 1500 kcal/day, and of a physical activity program [11]. The prescribed diet was based on Mediterranean principles, with a wide consumption of plant foods, cereals, legumes, and fish, with olive oil as the main source of fat, and moderate to no consumption of red wine. The dietary plan consisted of three main meals and three snacks divided as follows: breakfast, snack, lunch, snack, dinner, and evening snack before bedtime. For each meal or snack, several alternatives were offered to the pregnant women, all of which were suitably calibrated. The diet had a target macronutrient composition of 55% carbohydrates (80% complex carbohydrates with a low glycemic index and 20% simple carbohydrates), 20% protein (50% animal and 50% vegetable) and 25% fat (12% mono-unsaturated, 7% polyunsaturated, and 6% saturated) with moderately low saturated fat levels. To avoid ketonuria and acidosis, which frequently occur because of prolonged fasting, the daily recommended calories were divided into small, frequent meals. The daily intake of carbohydrates was at least 225 g/day, which is sufficient to prevent ketosis [12]. The primary focus of the dietary intervention was decreasing the consumption of foods with a high glycemic index and a high saturated fat content by substituting them with healthier alternatives based on the taste and preferences of the women and considering the country of origin.

Furthermore, the LI included a physical activity intervention, consistent with ACOG [13] and American College of Sport Medicine [14] recommendations for pregnant women. Women were advised to participate in 30 min of moderate-intensity activity at least three times per week.

The visits for women in the LI were scheduled at the 16th, 20th, 28th and 36th weeks of pregnancy with both the gynecologist and the dietitian. The compliance to LI was monitored at each visit. In addition to weight control, women were interviewed by the dietitian about the problems encountered with diet and physical habits prescriptions, and then counseled about possible changes when necessary.

### 2.2. Standard Care

Women in the SC group were routinely followed by the family centers or private practitioners at 10th–12th, 20th–22nd, and 34th weeks of pregnancy, and they performed an ultrasound, weight and health status check, and provided general recommendations on a healthy lifestyle in pregnancy in the form of simple and intuitive booklets created according to the recommendations of the Italian Society of Human Nutrition on the reference intake levels of nutrients and energy [10].

### 2.3. Data Collection

For all participating women, the gestational weight gain (GWG) was the difference between the initial weight and the final weight measured at the time of the last check-up in the clinic at 36th–37th weeks of gestation, when the mother came to open the medical record. In cases of preterm birth, the last weight measurement before delivery was considered. The GWG was defined as insufficient, adequate, or excessive according to IOM ranges [1]. The limits for weight gain were defined by the IOM (2 kg in the first trimester plus 0.27 kg/week in the second and third trimester for obese women) [15].

The data regarding the maternal and fetal outcomes were collected from the clinical records by two residents. In particular, the onset of Gestational Diabetes Mellitus (GDM), defined as a fasting blood sugar between 92 and 126 mg/dL or one or more values ≥ than the threshold values for OGTT 75 g performed between 24 and 28 weeks [16], the onset of gestational hypertension, defined as a blood pressure ≥ 140 mmHg systolic or 90 mmHg diastolic on 2 separate occasions at least 4 h apart after 20 weeks of pregnancy [17], the gestational age at delivery, the rate of cesarean section, birth weight, Apgar score at the 5th minute, need for resuscitation, and admission to a neonatal intensive care unit (NICU) were recorded. The birth weight centile was calculated in relation to the gestational age according to a large Italian study on neonatal anthropometrics [18]. The infants considered large for gestational age (LGA) were those with a birth weight ≥ the 90th centile, while the small-for-gestational-age (SGA) infants were those whose birth weights were ≤ the 10th centile.

### 2.4. Statistical Analysis

Data were analyzed using the statistical package Stata 16.1 (StataCorp, 2019, College Station, TX, USA). Continuous variables are reported as mean ± standard deviation (SD). Categorical variables are expressed as frequencies and percentages. After the normal distribution of the continuous variables was confirmed by the Shapiro–Wilk test, parametric tests were performed. All probability values were 2-tailed, and a probability value of <0.05 was considered statistically significant. The comparisons between the “lifestyle intervention” and “standard care”, were made using the t-student test for continuous variables and the chi-square test for categorical ones.

The variables that were considered clinically relevant in determining the LGA and SGA infants, with a *p*-value < 0.10 in the univariate analyses, were included in a multivariable logistic model. The final multivariate model was determined by a stepwise backward selection procedure in which only independent variables associated with SGA or to GDM with a *p*-value < 0.05 were retained. Results of logistic models were reported as the Odds Ratio (OR) with a 95% confidence interval and Wald *p*-value.

## 3. Results

A total of 1874 singleton obese women who delivered in the study period; among them, 624 were registered in the LI program. However, 59 (9%) dropped out, leaving 565 (30%) included in the LI program while 1309 (70%) received SC treatment. The main maternal sociodemographic variables collected at baseline and obstetric features are reported in Table 1.

The rate of nulliparous, Italian nationality, and obesity classes II and III were higher in the LI than SC group. Moreover, GWG was found more frequently insufficient in the SC than LI group.

Stratifying by obesity classes, insufficient GWG was found more frequently in obese class I (N = 614, 45.1%) with respect to class II (N = 2, 0.5%) and III (N = 4, 3.1%) (Table 2).

Women of the LI group showed a lower rate of SGA with respect to those of the SC group, whereas the LGA rate was similar between groups (Table 3). In women following the LI, there was a higher rate of gestational diabetes and hypertension with respect to the SC group. Labor and delivery outcomes were similar between groups (Table 3).

In the multivariate logistic regression, once accounting for maternal age, BMI classes, country of origin, care provider, and gestational hypertension, the risk for SGA was increased by insufficient GWG (OR = 1.25, 95%CI: 1.03–1.59), while it was reduced by LI (OR = 0.67; 95% CI: 0.42–0.98) (Table 4).

Moreover, another multivariate logistic regression demonstrated that the likelihood of developing GDM increases with the age in women with low education and decreases in Italian-born women with adequate GWG, once accounting for nulliparous status and obesity class (Table 5).

## 4. Discussion

In 2009, the IOM revised the guidelines for gestational weight gain, and the most important change was to recommend to obese women to gain less than overweight women. The impact of these guidelines on the US population was a reduction in preterm labor and low- and very-low birthweight, without affecting the rate of adequate GWG and that of gestational diabetes [19]. These data could be explained considering the low adherence to the guidelines, especially among women of low socioeconomic status with a more severe obesity reporting a negative healthcare experience that leads to the avoidance or delay of care [20].

The present study was performed in a different population with respect to the US one, characterized by a universal health system; thus, the subjects received antenatal care irrespective of socioeconomic status. In our cohort, when a lifestyle intervention based on a low-caloric diet and scheduled physical activity was implemented, this allowed more women to have adequate GWG, reducing the extremes (either excessive or insufficient). This result could be explained by the fact that women belonging to an upper socioeconomic status and with a higher degree of obesity are more aware of the pregnancy risks related to their condition.

As a consequence, the intervention did not significantly impact LGA babies, while it was associated with a lower rate of SGA newborns. As previously reported, reduced fetal growth could be due to an insufficient maternal weight gain in obese women [7,8], a risk factor which was independently associated with SGA in our series, too. Interestingly, participation in an LI program was able to counteract such a negative outcome.

It has to be highlighted that women referred for LI differ from those receiving standard advice about lifestyle changes. While these differences were accounted for in the multivariate analysis, leaving the conclusions credible, the issue of translating these results to the general population remains.

Indeed, systematic reviews and meta-analyses, even using sensitive methods such as Individual Patient Data, were unable to conclude on the effect of lifestyle interventions on neonatal outcomes [21,22,23]. The reasons for such a failure despite expectations are several, including diverse interventions, different populations, and failure to check the adherence to the prescribed diet and/or physical activity.

Socioeconomic status remains a discriminant factor, as demonstrated by the lower rate of non-Italian women in the LI group. In view of this, interventions to improve health policy in different ethnic groups should be implemented.

The obesity degree also influences adherence to lifestyle interventions; obese women, independently of the obesity classes, should be informed about the risk of an excessive or insufficient GWG and should be directed to a specific lifestyle path during pregnancy. Recommendations about the GWG for each obesity class, lacking in the IOM guidelines, could improve the medical approach of these women in order to improve maternal and fetal outcomes.

In this regard, a very recent population-based cohort study published in the Lancet [24] confirmed that in pregnancies with class I or II obesity, gestational weight gain values below the lower limit of the IOM recommendation or weight loss did not increase the risk of adverse composite outcomes (i.e., stillbirth, infant death, LGA, SGA, preterm birth, unplanned cesarean delivery, gestational diabetes, pre-eclampsia, etc.), and in pregnancies with class III obesity, were associated with a reduced risk of adverse composite outcomes.

This supports the need to lower or remove the lower limit of the current IOM recommendations for pregnant women with obesity and suggests that separate guidelines for class III obesity may be warranted.

However, it is worth mentioning that many are the factors contributing to SGA newborns and they are not solely related to maternal body weight and nutrition.

Placental functionality, for instance, is one of the main drivers that should be considered. Indeed, it is well known that maternal obesity is characterized by a chronic low-grade inflammation that is the trigger of impaired placental development and function. Placental angiogenesis and trophoblast invasion are indeed altered by inflammatory cytokines, leading to fetal growth restriction and other adverse pregnancy outcomes [25]. Therefore, considering the growing evidence that supports the hypothesis that dietary factors may play a role in reducing systemic low-grade chronic inflammation [26], it becomes crucial to implement such a low-risk and health promoting strategy in such a vulnerable population as represented by obese pregnant women.

The strength of this study is that it sheds light on the impact that a lifestyle program can have on pregnancy outcomes in each class of obese women, considering the limited availability of Italian data on this topic. Moreover, a small percentage of lost to follow up was found between the beginning of the study and the collection of pregnancy and perinatal outcomes.

However, an important limitation of the present study is based on the fact that mothers who agreed to participate in the study may be those with better results because they showed a more positive attitude towards the application of the lifestyle intervention and more likely adhered to the diet and physical activity program. Therefore, it is not possible to confirm that this sample is representative of the baseline population of obese women. Furthermore, the research team could not guarantee that the women included in the SC group, followed by family centers or private practitioners, did not individually follow any other lifestyle interventions provided by other private dietitians or personal trainers. However, this is a very remote possibility, so no impact on the study’s main outcome is estimated.

Another limitation worth recognizing is related to the lack of information on dietary intake in both groups and on the use of supplements. In fact, the exact intake of nutrients, together with the use of food supplements, widely consumed by pregnant women, was not reported in the study database and therefore was not analyzed, although it was investigated during the interview with the women of the LI group. This represents a methodological flaw.

## 5. Conclusions

In conclusion, in this prospective cohort study on obese pregnant women, participation in a controlled LI was associated with more adequate GWG, reduced insufficient weight gain, and thus decreased risk of SGA infants. Therefore, professionals who follow obese women during pregnancy should give importance to the introduction of lifestyle interventions aimed at improving the intake of nutrients, not necessarily increasing them, but improving their quality through the nutritional and physical re-education of obese women during pregnancy.

Further studies are needed to thoroughly investigate the impact on the metabolic changes linked to this LI on obese pregnant women, taking into higher consideration the exact intake of nutrients and the eventual impact of food supplementation.

## Figures and Tables

**Table 1 nutrients-16-03261-t001:** Maternal baseline and pregnancy features.

		Control (N = 1309)	Lifestyle Intervention (N = 565)	*p*-Value
Mean maternal age (years)		36.5 ± 5.6	34.8 ± 5.7	0.000
Maternal age ≥ 40		401 (30.6)	151 (26.7)	0.09
Nulliparity		434 (33.1)	213 (37.7)	0.05
Parity	1	599 (45.8)	201 (35.5)	0.002
	2	180 (13.7)	95 (16.8)	
	3	96 (7.4)	56 (9.9)	
Nationality				0.000
Italy		709 (54.1)	402 (71.1)	
Sub-Saharan Africa		251 (19.1)	83 (14.7)	
Magreb		290 (22.1)	60 (10.6)	
Others		59 (4.5)	20 (3.5)	
Low education (≤8 years)		673 (51.4)	279 (49.4)	0.37
Obesity classes				0.001
Class I		971 (74.2)	373 (66.0)	
Class II		257 (19.6)	144 (25.5)	
Class III		81 (6.2)	48 (8.5)	
Mean pre-pregnancy BMI		33.4 ± 3.3	34.2 ± 4.3	0.000
Mean GA at start of intervention (weeks)		11 ± 0.3	11 ± 0.8	0.87
Care provider				0.02
Public (family centers)		960 (73.4)	449 (79.5)	
Private (gynecologist)		349 (26.6)	116 (20.5)	

**Table 2 nutrients-16-03261-t002:** Effects of LI on pregnancy diseases and GWG.

	Control (N = 1309)	Lifestyle Intervention (N = 565)	*p* Value
Gestational Diabetes Mellitus	401 (30.6)	202 (35.7)	0.03
Gestational hypertension	89 (6.8)	58 (10.3)	0.001
Excessive GWG (≥9 kg)	361 (27.6)	133 (23.5)	0.04
Adequate GWG (5–9 kg)	512 (39.1)	272 (48.1)	0.001
Insufficient GWG (<5 kg)	436 (33.3)	160 (28.3)	0.01

**Table 3 nutrients-16-03261-t003:** Labor, delivery, and perinatal outcomes.

	Control (N = 1309)	Lifestyle Intervention (N = 565)	*p*-Value
Labor induction	481 (36.7)	208 (36.8)	0.47
Caesarean section	377 (28.8)	162 (28.7)	0.49
Operative delivery	67 (5.1)	21 (3.7)	0.09
Preterm delivery (<37 weeks)	96 (7.3)	47 (8.3)	0.41
Birthweight	3333.5 ± 579.4	3323.2 ± 602.3	0.72
LGA	265 (20.2)	105 (18.6)	0.18
SGA	98 (7.5)	29 (5.1)	0.03

**Table 4 nutrients-16-03261-t004:** Multivariate logistic regression for the likelihood of having an SGA newborn.

	OR	95% CI	*p*-Value
Obesity classes *§ Class I* *Class II* *Class III*	1.24	0.90–1.71	0.17
Italian place of origin	1.19	0.83–1.73	0.33
Lifestyle intervention	0.67	0.42–0.98	0.05
Maternal age	1.00	0.97–1.03	0.85
Public cares provider	0.94	0.61–1.45	0.80
Insufficient GWG	1.25	1.03–1.59	0.04
Gestational hypertension	1.03	0.52–2.07	0.87

OR: Odds Ratio, §: reference group.

**Table 5 nutrients-16-03261-t005:** Multivariate analysis to evaluate the variables impacting the likelihood of developing GDM.

	OR	95% CI	*p*-Value
Low Education	1.24	1.00–1.55	0.04
Italian place of origin	0.72	0.57–0.90	0.004
Nulliparity	0.91	0.72–1.14	0.43
Maternal Age	1.05	1.03–1.10	0.000
Class III Obesity	1.03	0.85–1.25	0.72
Adequate GWG	0.96	0.94–0.98	0.001

## Data Availability

The data that support the findings of this study are not publicly available due to the privacy of research participants, but are available from the corresponding author D.M. upon reasonable request.

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
