# Peer review of "Lifestyle Intervention to Promote an Adequate Gestational Weight Gain and Improve Perinatal Outcomes in a Cohort of Obese Women"

_nutrients, 2024, doi:10.3390/nu16193261_

Round 1

Reviewer 1 Report

Comments and Suggestions for Authors

The authors have looked into an important aspect of managing obese pregnant women. This data is interesting to examine as the authors aim to include ethnic specific and socio-economic status on GWG in women above BMI 30. The findings are relevant to clinical practice and it will be ideal to share an example of the personalized lifestyle intervention programme in the methods section. Additionally, the outcome which is having the most appropriate GWG reduces the risk of SGA. However, the incidence of gestational hypertension and other risk factors seem to be in higher proportion in the LI group - can the authors clarify if there are any analyses which demonstrate that LI with the appropriate GWG actually reduces the risks and improves outcomes as well such as SGA, gestational hypertension and preterm deliveries in this group? It would be ideal to show this data and this further reduction and improvement of outcomes.

Comments on the Quality of English Language

There are some spelling and grammatical errors throughout the manuscript which need to be corrected and to ensure clarity of expression in the English language.

Author Response

Reviewer #1

Comments 1: The authors have looked into an important aspect of managing obese pregnant women. This data is interesting to examine as the authors aim to include ethnic specific and socio-economic status on GWG in women above BMI 30. The findings are relevant to clinical practice and it will be ideal to share an example of the personalized lifestyle intervention programme in the methods section. Additionally, the outcome which is having the most appropriate GWG reduces the risk of SGA. However, the incidence of gestational hypertension and other risk factors seem to be in higher proportion in the LI group - can the authors clarify if there are any analyses which demonstrate that LI with the appropriate GWG actually reduces the risks and improves outcomes as well such as SGA, gestational hypertension and preterm deliveries in this group? It would be ideal to show this data and this further reduction and improvement of outcomes.

Responses 1: We thank the reviewer for the comments and suggestions. Regarding the possibility of including an example of the personalized lifestyle intervention it would be very hard for the authors to summarize it in the material section, as the lifestyle intervention is tailored on each single women, according to her eating habits, tastes, religion, and so on. Therefore, we have provided the generic indications on what is based the programme which are included in the methods section.

We acknowledge the higher rate of Gestational Hypertension in the LI group, probably related to the higher rate of class II and class III obese women in such group.

However, by including this variable in the Multivariate logistic regressions (tables 4 and 5), we have provided a result on the effect of Lifestyle intervention and of GWG, independently from the different incidence of gestational hypertension among the two groups.

Reviewer 2 Report

Comments and Suggestions for Authors

Journal: nutrients    Manuscript ID: nutrients-3185993

Title: "Lifestyle intervention to promote an adequate gestational weight gain and improve perinatal outcomes in a cohort of obese women"

Author: Daniela Menichiniet al.

The authors in the present study explored the association of gestational weight gain (GWG) with pregnancy and perinatal outcomes among 1,874 obese singleton women receiving standard care (SC) or a lifestyle intervention (LI) including a low-calorie diet and physical activity. According to the study findings, obese women of the LI group had higher rates of adequate GWG, while women of the SC group showed a higher incidence of inadequate or excessive GWG and an increased rate of small-for-gestational-age (SGA) neonates. The study findings are interesting and add clinical insights. However, the following points merit consideration.

Comments:

1.     Please ensure that all abbreviations are defined the first time they are mentioned in the main text and in the abstract (e.g., SGA).

2.     In the text, the authors use "Obese women class II-III" and "obese class 3." For consistency in terminology, I would suggest using one description throughout.

3.     Is the “<10° and <3° percentiles” notation in the manuscript meant to be “<10th and <3rd percentiles”?

4.     Please review the manuscript for grammatical errors or typos (e.g., “kg/m2”).

5.     The authors should further clarify how each individual was enrolled to follow each strategy (standard care or lifestyle intervention), provide details regarding how physicians and patients were involved in this decision, and explain the referral process for pregnant women to the Obesity weight-management ad-hoc clinic.

6.     In the “Materials and Methods” section, please include the inclusion and exclusion criteria of the study.

7.     Define “the last few weeks prior to delivery”. Specify the time of assessment for each group. How did the authors ensure that differences in the timing of assessments between the two groups did not impact the study’s outcomes?

8.     What does the "general recommendation on a healthy lifestyle" include? Please specify.

9.     Did the authors assess the normal distribution of the variables before performing parametric tests?

10.  Please provide further information on how "interaction was verified" and which interactions were assessed.

11.  In Table 1, please include age as a continuous variable, socioeconomic status, number of previous births, marital status, presence of Type 2 Diabetes (T2D) or a history of prediabetes, blood pressure, glucose levels, A1c values, the gestational week when the intervention started.

12.  Since Table 2 includes variables besides GWG classes, I suggest modifying the title accordingly. Also, specify how long each intervention was followed by each pregnant woman until the Gestational Diabetes Mellitus and Gestational Hypertension variables were assessed.

13.  Please clarify in the Materials and Methods section how Gestational Diabetes Mellitus (GDM) was defined.

14.  The authors should provide information (also in a table comparing the two groups) regarding the dietary information (e.g., macronutrients, fiber, etc.) and physical activity followed by the participants in each group.

15.  How was compliance and adherence determined? Additionally, how did the authors ensure that the women followed by family centers or private settings representing the SC group did not follow lifestyle interventions?

16.  Please provide a flow chart. Also explain the reasons for participants dropping out.

17.  Please ensure that all potential limitations associated with the current study and the inherent limitations of the study design are described in the relevant paragraph.

Author Response

Reviewer #2

Journal: nutrients    Manuscript ID: nutrients-3185993

Title: "Lifestyle intervention to promote an adequate gestational weight gain and improve perinatal outcomes in a cohort of obese women"

Author: Daniela Menichini et al.

The authors in the present study explored the association of gestational weight gain (GWG) with pregnancy and perinatal outcomes among 1,874 obese singleton women receiving standard care (SC) or a lifestyle intervention (LI) including a low-calorie diet and physical activity. According to the study findings, obese women of the LI group had higher rates of adequate GWG, while women of the SC group showed a higher incidence of inadequate or excessive GWG and an increased rate of small-for-gestational-age (SGA) neonates. The study findings are interesting and add clinical insights. However, the following points merit consideration.

 Comment 1.  Please ensure that all abbreviations are defined the first time they are mentioned in the main text and in the abstract (e.g., SGA).

Response 1. We thank the reviewer. We have defined all abbreviations the first time they are mentioned also in the abstract. Changes implemented are visible in track change mode directly in the manuscript.

Comment 2.  In the text, the authors use "Obese women class II-III" and "obese class 3." For consistency in terminology, I would suggest using one description throughout.

Response 2. We Agree. The manuscript was homogenised by using always class II-III, please see the track changes version.

Comment 3. Is the “<10° and <3° percentiles” notation in the manuscript meant to be “<10th and <3rd percentiles”?

Response 3. Yes, we confirm that “<10° and <3° percentiles” notation was meant to be “<10th and <3rd percentiles”, we have corrected in the manuscript.

Comment 4. Please review the manuscript for grammatical errors or typos (e.g., “kg/m2”).

Response 4. We thank the reviewer for the comment. We have performed an extensive grammar check through the text and corrected grammatical errors and typos, highlighted in the track change version.

Comment 5. The authors should further clarify how each individual was enrolled to follow each strategy (standard care or lifestyle intervention), provide details regarding how physicians and patients were involved in this decision, and explain the referral process for pregnant women to the Obesity weight-management ad-hoc clinic.

Response 5. We agree with the reviewer. Please see the following paragraph, which was rephrased to better clarify the referral process: “The study included singleton pregnant women with a pre-pregnancy BMI ≥ 30 kg/m2 and age ≥ 18 years, delivered from 2016 and 2020 at the Policlinic Hospital of Modena. Among them, the family centers physicians and midwives proposed to a group of women the possibility to be referred to an Obesity weight-management ad-hoc clinic at the Obstetric Unit at the Mother-Infant Department. The women that agreed, represented the LI group, and were exposed, starting from their 9th-12th weeks of pregnancy, to a personalized dietary intervention tailored on their eating habits, taste and religion, with extensive explanation of meal subdivision and possible food substitutions and to a physical activity program.”

Comment 6. In the “Materials and Methods” section, please include the inclusion and exclusion criteria of the study.

Response 6. We thank the reviewer for the suggestion. The following paragraph reporting the inclusion and exclusion criteria was added.

“The inclusion criteria for the study were: having a BMI ≥ 30 kg/m2, age ≥ 18 years, Sin-gle pregnancy, GA ≤ less than 12 weeks, willingness to participate in the study. The exclusion criteria were mainly represented by the presence of chronic diseases, in particular pregestational hypertension, type 1 or 2 diabetes mellitus, cardiovascular pathologies and kidney disease that contraindicate caloric restriction or physical activity, thyroid pathologies in poor compensation, inability to follow the diet prescribed for cultural, ethnicity or religion-linked reasons.”

Please see the track change version of the manuscript.

Comment 7. Define “the last few weeks prior to delivery”. Specify the time of assessment for each group. How did the authors ensure that differences in the timing of assessments between the two groups did not impact the study’s outcomes?

Response 7. We thank the reviewer for the fair comment. We have better specified what we meant by saying “the last few weeks prior to delivery” and how the measurement of weight was standardised among participants.

“For all participating women, the gestational weight gain (GWG) was the difference between the initial weight and a final weight taken measured at the time of the last check-up in the clinic, at 36th – 37th weeks of gestation, when the mother came to open the medical record. In cases of preterm birth, the last weight measurement before de-livery was considered.”

Please see the track change version of the manuscript.

Comment 8.  What does the "general recommendation on a healthy lifestyle" include? Please specify.

Response 8. We thank the reviewer for the comment. We have specified in the text that the recommendation on healthy lifestyle were provided “in the form of simple and intuitive booklets created according to the recommendations of the Italian Society of Human Nutrition on reference intake levels of nutrients and energy [10].”

Comment 9.  Did the authors assess the normal distribution of the variables before performing parametric tests?

Response 9. Yes, the normality check was performed before using the parametric tests. We have specified this in the statistical analysis paragraph. Please see the track change version of the manuscript.

Comment 10. Please provide further information on how "interaction was verified" and which interactions were assessed.

Response 10. The sentence was not clear, therefore we decided to eliminate it as no major interactions were present among the studied variables. We apologize for the oversight.

Comment 11.  In Table 1, please include age as a continuous variable, socioeconomic status, number of previous births, marital status, presence of Type 2 Diabetes (T2D) or a history of prediabetes, blood pressure, glucose levels, A1c values, the gestational week when the intervention started.

Response 11. We thank the reviewer for the comment. The maternal age as continuous variable, the number of previous birth and the mean gestational age at the start of intervention were added to table 1. The presence of Type 2 Diabetes (T2D) or a history of prediabetes are not applicable as the T2DM or the history of prediabetes were considered exclusion criteria. The other requested variables blood pressure, glucose levels, A1c values were not reported in the study database.

Comment 12.  Since Table 2 includes variables besides GWG classes, I suggest modifying the title accordingly. Also, specify how long each intervention was followed by each pregnant woman until the Gestational Diabetes Mellitus and Gestational Hypertension variables were assessed.

Response 12. We agree with the suggestion. The title of table 2 was changed into “Effects of LI on pregnancy diseases and GWG”.

Regarding how long each intervention was followed by each pregnant woman until the Gestational Diabetes Mellitus and Gestational Hypertension variables were assessed, the time was the same as women were included since 9-12 weeks and the diagnosis of GDM and GH are performed generally around the same gestational age, 24-28 weeks and after 20th week, respectively. We do not have the exact diagnosis date to calculate this datum for each woman.

Comment 13.  Please clarify in the Materials and Methods section how Gestational Diabetes Mellitus (GDM) was defined.

Response 13. We agree, the definition of Gestational Diabetes Mellitus and Gestational Hypertension were provided in the Materials and Methods section, with relative references.

Comment 14.  The authors should provide information (also in a table comparing the two groups) regarding the dietary information (e.g., macronutrients, fiber, etc.) and physical activity followed by the participants in each group.

Response 14. We appreciate the interesting comment, however such comparison cannot be performed, as the SC group was not prescribed a detailed diet, and the dietary intake was not collected. We will take into consideration this comment for future studies.

Comment 15.  How was compliance and adherence determined? Additionally, how did the authors ensure that the women followed by family centers or private settings representing the SC group did not follow lifestyle interventions?

Response 15. A sentence was added in the Material and Methods section to better detail how that compliance was monitored. “The compliance to LI interventions was monitored at each visit. In addition to weight control, women were interviewed by the dietitian about the problems encountered with diet and physical habits prescriptions, then counselled about possible changes when necessary.”

The research team could not ensure that the women included in the SC group, followed by family centers or private settings, did not individually follow any other lifestyle intervention by other private dietician. However, this seems unlikely, considering that women of family centers at the least has no propension to private, money costing intervention. As far as women choosing a private obstetrician provider, we expect their % would small given the still poor sensitivity of doctors and patients respect with obesity in pregnancy. Anyway, we will add this as a limitation of the study.

Comment 16.  Please provide a flow chart. Also explain the reasons for participants dropping out.

Response 16. We thank the reviewer for the useful comment. The dropouts have been defined for the LI group and were considered those women that did not present to planned clinical check more than two times. Accordingly, their rate is   9.7 %.

A sentence was added in the result section – line 160: “624 were registered in LI program. However, 59 dropped out leaving 565 (30%) ….”

This study is not a RCT and reasons for dropping-out were not monitored.

Comment 17.  Please ensure that all potential limitations associated with the current study and the inherent limitations of the study design are described in the relevant paragraph.

Response 17. We thank the reviewer and confirm that the additional limitations highlighted during the revision process were added to the relevant paragraph.

Reviewer 3 Report

Comments and Suggestions for Authors

The study was conducted on a very large group of patients, which is a significant advantage.

1. The introduction would benefit from a paragraph on adverse metabolic changes in obese women and the more frequent occurrence of metabolic diseases, such as diabetes, lipid and liver profile disorders.

A few methodological comments would help to explain the profile of the study.

2. It would be beneficial to understand how the remaining nutrients were balanced, including vitamins, minerals and fibre.

3. It would be beneficial to ascertain whether the women received any form of supplementation, and if so, what form it took.

4. Please describe the method used to monitor compliance with dietary guidelines.

5. At what gestational age were the women included in the study?

6. Please describe the inclusion criteria for the study.

7. Furthermore, no anthropometric measurements were taken of the study group. It is therefore unclear whether the average refers to the woman's pre-pregnancy measurements, her measurements at the time of inclusion in the study after pregnancy was confirmed, or her measurements after the end of pregnancy.

8. It is unclear how many women had miscarriages and whether they were excluded from the study.

10. The discussion should take into account the fact that obese women are more susceptible to preeclampsia, which may contribute to SGA.

https://doi.org/10.3390/ijms21249628

11. Furthermore, inflammation may impede the proper anchoring of the placenta

https://doi.org/10.3390/jcm12185995

12. The increase in body weight is not the sole factor contributing to the low body weight of newborns.

Conclusion

13. It is recommended that further consideration be given to the introduction of dietary restrictions in obese women during pregnancy.

Best regards

Comments on the Quality of English Language

I am not qualified to judge the quality of the English language in this article, but I had no difficulty in understanding the article.

Author Response

Reviewer #3

The study was conducted on a very large group of patients, which is a significant advantage.

Comment 1. The introduction would benefit from a paragraph on adverse metabolic changes in obese women and the more frequent occurrence of metabolic diseases, such as diabetes, lipid and liver profile disorders.

A few methodological comments would help to explain the profile of the study.

Response 1. We thank the reviewer for the comments. A brief general paragraph on the impact of metabolic alteration related to obesity in pregnancy was added in the introduction.

“Obesity has become a serious health concern worldwide. The metabolic changes, the pro-inflammatory chronic status that characterize obesity are associated with several health complications and adverse outcomes for both the mother and the fetus, such as preeclampsia, gestational diabetes, preterm birth, and fetal growth restriction.”

Comment 2. It would be beneficial to understand how the remaining nutrients were balanced, including vitamins, minerals and fibre.

Response 2. We thank the reviewer for the comments and suggestions. Regarding the possibility of detailing how the nutrients were balanced, including vitamins, minerals and fibre, it would be very hard for the authors to summarize it in the material section, as the lifestyle intervention is tailored on each single women, according to her eating habits, tastes, religion, and so on. Therefore, we have provided the generic indications on what is based the programme which are included in the methods section.

Comment 3. It would be beneficial to ascertain whether the women received any form of supplementation, and if so, what form it took.

Response 3. We appreciate the useful comment, however no data was collected on the supplementations that women took during pregnancy. We therefore cannot further detail this aspect; we will however consider including this information in future studies.

Comment 4. Please describe the method used to monitor compliance with dietary guidelines.

Response 4.  A sentence was added in the Material and Methods section to better detail how that compliance was monitored. “The compliance to LI interventions was monitored at each visit. In addition to weight control, women were interviewed by the dietitian about the problems encountered with diet and physical habits prescriptions, then counselled about possible changes when necessary.”

Comment 5. At what gestational age were the women included in the study?

Response 5. Women were included between 9-12th weeks of GA. Further details on the mean GA at the start of intervention have been included in table 1.

Comment 6. Please describe the inclusion criteria for the study.

Response 6. A paragraph reporting inclusion and exclusion criteria was included in the Material and Methods section. Please see the track changes version of the Manuscript.

Comment 7. Furthermore, no anthropometric measurements were taken of the study group. It is therefore unclear whether the average refers to the woman's pre-pregnancy measurements, her measurements at the time of inclusion in the study after pregnancy was confirmed, or her measurements after the end of pregnancy.

Response 7. We thank the reviewer for the comment. We clarify that the mean BMI reported in table 1 is the pre-pregnancy measurement. Subsequently, the weight and BMI were taken either at the study inclusion (9-12 weeks) at each visit during pregnancy, and before delivery (at 36-37th weeks of pregnancy).

Comment 8. It is unclear how many women had miscarriages and whether they were excluded from the study.

Response 8. Women with miscarriage were not reported in the manuscript, as only women delivering at the Policlinic Hospital of Modena were finally considered for the analysis.

Comment 9. The discussion should take into account the fact that obese women are more susceptible to preeclampsia, which may contribute to SGA.

https://doi.org/10.3390/ijms21249628

Response 9. We really appreciate the reviewer’s comment. A paragraph was added in the discussion section on the other factors contributing to an SGA newborn. “However, it is worth mentioning that many are the factors contributing to the SGA newborns and are not solely related to maternal body weight and nutrition.

Placental functionality, for instance, is one of the main drivers that should be taken into account. Indeed, it is well known that maternal obesity is associated with chronic low-grade inflammation, which can lead to impaired placental development and function. Inflammatory cytokines can disrupt trophoblast invasion and placental angiogenesis, leading to fetal growth restriction and other adverse pregnancy outcomes [25]. Therefore, considering the growing evidence that supports the hypothesis that dietary factors may play a role in reducing the systemic low-grade chronic inflammation [26], it becomes crucial to implement such a low-risk and health promoting strategy on such a vulnerable population represented by obese pregnant women.”

Comment 10. Furthermore, inflammation may impede the proper anchoring of the placenta

https://doi.org/10.3390/jcm12185995

Response 10. Please see the response 9, which included the comment 9, 10 and 11.

Comment 11. The increase in body weight is not the sole factor contributing to the low body weight of newborns.

Response 11. Please see the response 9, which included the comment 9, 10 and 11.

Conclusion

Comment 12. It is recommended that further consideration be given to the introduction of dietary restrictions in obese women during pregnancy.

Response 12. We thank the reviewer for the comment. Further emphasis was given in the conclusive paragraph on the introduction of lifestyle intervention program for such population. “Therefore, professionals who follow obese women during pregnancy, should give importance to the introduction of lifestyle interventions aimed at improving the intake of nutrients, not necessarily limiting them, but improving their quality through nutritional and physical re-education of women obese during pregnancy.”

Round 2

Reviewer 2 Report

Comments and Suggestions for Authors

Journal: nutrients            Manuscript ID: nutrients-3185993 (Revised version)

Title: "Lifestyle intervention to promote an adequate gestational weight gain and improve perinatal outcomes in a cohort of obese women"

Author: Daniela Menichiniet al.

The authors have tried to address my comments and made the appropriate revisions, further improving the manuscript. However, there are a few minor comments for the revised manuscript that still need to be addressed:

1.     The authors in their response regarding how they assessed the normal distribution did not specify which steps they have taken; it is also not mentioned in the statistical analysis section. Could you please clarify this point?

2.     Regarding the issue about how “interaction was verified”, the authors responded that "The sentence was not clear, therefore we decided to eliminate it as no major interactions were present among the studied variables." However, the statement “interaction was verified” still exists in the statistical section. Please elaborate on this.

3.     Please adjust the models to account for maternal age as a continuous variable, since a significant difference is observed between the two groups. Similarly, adjust for the parity variable.

4.     The authors should acknowledge the lack of dietary intake information as a limitation.

Author Response

Reviewer#2

Journal: nutrients            Manuscript ID: nutrients-3185993 (Revised version)

Title: "Lifestyle intervention to promote an adequate gestational weight gain and improve perinatal outcomes in a cohort of obese women"

Author: Daniela Menichini et al.

The authors have tried to address my comments and made the appropriate revisions, further improving the manuscript. However, there are a few minor comments for the revised manuscript that still need to be addressed:

Comment 1. The authors in their response regarding how they assessed the normal distribution did not specify which steps they have taken; it is also not mentioned in the statistical analysis section. Could you please clarify this point?

Response 1. We thank the reviewer for the comment. We specified in the statistical Analysis paragraph what test was used to assess the normality of continuous variables. Please see amended text below:

“After the normal distribution of the continuous variables was confirmed by the Shapiro–Wilk test, the parametric tests were performed.”

Comment 2. Regarding the issue about how “interaction was verified”, the authors responded that "The sentence was not clear, therefore we decided to eliminate it as no major interactions were present among the studied variables." However, the statement “interaction was verified” still exists in the statistical section. Please elaborate on this.

Response 2. The authors apologise for the oversight. The statement “interaction was verified” was removed from the main text.

Comment 3.  Please adjust the models to account for maternal age as a continuous variable, since a significant difference is observed between the two groups. Similarly, adjust for the parity variable.

Response 3. We thank the reviewer for the comment. As suggested, the Maternal age (year) as a continuous variable was included in the models (table 4 and 5). No relevant changes were found in the statistically significant variables impacting on the dependent one except for the the impact “Low education” which became statistically significant on determining GDM, accounting for all the other variables included in the model reported in table 5.

Table 4. Multivariate logistic regression for the likelihood of having a SGA newborn.

OR

95% CI

P value

Obesity classes

§ Class I

Class II

Class III

1.24

0.90 – 1.71

0.17

Italian place of origin

1.19

0.83 – 1.73

0.33

Lifestyle intervention

0.67

0.42 – 0.98

0.05

Maternal age

1.00

0.97 – 1.03

0.85

Public cares provider

0.94

0.61 – 1.45

0.80

Insufficient GWG

1.25

1.03 – 1.59

0.04

Gestational hypertension

1.03

0.52 – 2.07

0.87

Table 5. Multivariate analysis to evaluate the variables impacting on the likelihood of developing a GDM

OR

95% CI

P value

Low Education

1.24

1.00 – 1.55

0.04

Italian place of origin

0.72

0.57 – 0.90

0.004

Nulliparity

0.91

0.72 – 1.14

0.43

Maternal Age

1.05

1.03 – 1.10

0.000

Class III Obesity

1.03

0.85 – 1.25

0.72

Adequate GWG

0.96

0.94 – 0.98

0.001

Comment 4.     The authors should acknowledge the lack of dietary intake information as a limitation.

Response 4.  As suggested, we acknowledge this limitation in the manuscript. Please see text  below: “Another limitation worth recognizing is related to the lack of information on dietary intake in both groups and on the use of supplements. In fact, the exact intake of nutrients, together with the use of food supplements, widely consumed by pregnant women, was not reported in the study database and therefore was not analysed, although it was investigated during the interview with the women of the LI group. This represents a methodological flaw.”

Reviewer 3 Report

Comments and Suggestions for Authors

1. The authors did not address the reviewer's suggestions, some answers are insufficient, and the added paragraphs do not contain any citations.

2. It is evident that pregnant women receive supplements, and thus the omission of this information from the interview is a significant methodological flaw that may influence the conclusions drawn. The authors did not include this fact in the limitations of the study.

3. Furthermore, the precise point in time at which the data presented in Table 1 was collected remains unclear.

4. Any amendments to the manuscript should be clearly visible to the reviewer (e.g. in colour or bold).

Author Response

Reviewer #3

Comment 1. The authors did not address the reviewer's suggestions, some answers are insufficient, and the added paragraphs do not contain any citations.

Response 1. We really appreciate the reviewer’s comment and suggestion, however we preferred not to emphasize obesity-related metabolic changes much in the introductory paragraph, as our study does not report metabolic outcomes, but rather we included this point among the perspectives for future studies.

Further studies are needed to thoroughly investigate the impact on the metabolic changes linked to this LI on obese pregnant women, taking into higher consideration the exact intake of nutrients and the eventual impact of food supplementation.”

 Please note that the added paragraph in the introduction section does not contain citations as it was not referred to a specific work, rather to a general consideration made by the authors.

Comment 2. It is evident that pregnant women receive supplements, and thus the omission of this information from the interview is a significant methodological flaw that may influence the conclusions drawn. The authors did not include this fact in the limitations of the study.

Response 2.  As suggested, we acknowledge this limitation in the manuscript. Please see text  below: “Another limitation worth recognizing is related to the lack of information on dietary intake in both groups and on the use of supplements. In fact, the exact intake of nutrients, together with the use of food supplements, widely consumed by pregnant women, was not reported in the study database and therefore was not analysed, although it was investigated during the interview with the women of the LI group. This represents a methodological flaw.”

Comment 3. Furthermore, the precise point in time at which the data presented in Table 1 was collected remains unclear.

Response 3. We thank the reviewer for the comment. To better clarify the timepoint in which data reported in table 1 were collected, the manuscript was modified as reported below:

”A total of 1874 singleton obese women delivered in the study period, among them 624 were registered in LI program. However, 59 (9%) dropped out leaving 565 (30%) included in the LI program while 1309 (70%) received SC treatment. The main maternal sociodemographic variables collected at baseline and obstetric features are reported in Table 1.

Table 1. Maternal baseline and pregnancy features.”

Comment 4. Any amendments to the manuscript should be clearly visible to the reviewer (e.g. in colour or bold).

Response 4. We inform the reviewer that a track changes version of the Manuscript was attached to this and to the previous round of submission.
